# Carbonation Behavior of Mortar Made from Treated Recycled Aggregates: Influence of Diammonium Phosphate

**DOI:** 10.3390/ma16030980

**Published:** 2023-01-20

**Authors:** Diana Gómez-Cano, Yhan P. Arias-Jaramillo, Roberto Bernal-Correa, Jorge I. Tobón

**Affiliations:** 1Department of Construction, School of Architecture, Universidad Nacional de Colombia, Medellín 050034, Colombia; 2Orinoquia Institute of Studies, Universidad Nacional de Colombia, Arauca, Colombia; 3Department of Materials and Minerals, School of Mines, Universidad Nacional de Colombia, Medellín 050034, Colombia

**Keywords:** recycled concrete fine aggregates (RFA), DAP treatment, hydroxyapatite (HAP), recycle concrete (RC), durability, carbonation phenomenon, compression strength

## Abstract

This research aims to improve the quality of recycled concrete fine aggregates (RFA) by using diammonium hydrogen phosphate (DAP). We aimed to understand the effect of DAP treatment on durability performance due to the carbonation action of mortars with the partial and total substitution of treated RFA. The results showed a maximum reduction in the RFA water absorption of up to 33% using a minimum DAP concentration due to a pore refinement as a consequence of the formation of calcium phosphates such as hydroxyapatite (HAP). The carbonation phenomenon did not have a significant effect on the durability of mortars with DAP-treated RFA, as we did not find a decrease in the compressive strength; the carbonation depth of the mortars with 100% treated RFA decreased up to 90% and 63% for a w/c of 0.45 and 0.50, in comparison with mortars with 0% treated RFA. An inversely proportional relationship was found between the accelerate carbonation and the compressive strength, showing that higher percentages of treated RFAs in the mortar promoted an increase in compressive strength and a decrease in the carbonation rate, which is behavior associated with a lower permeability of the cement matrix as one of the consequences of the microstructural densification by DAP treatment.

## 1. Introduction

Globally, concrete is the substance used in the largest quantity by mass after water, with a foreseen increase in demand in the next decades [1]. The construction industry produces large amounts of construction and demolition waste (CDW) [2], with about 30 Gt/year of CDW, where China, European countries and the United States are the largest generators [3], and from which about 70% of the CDW is concrete [4,5]. More importantly, concrete production depends on natural resources, such as exhaustible aggregates (fine and coarse), as the main component of a conventional mix with about 80–85% of the total volume. Worldwide, the amount of aggregates extraction is around 26 Gt/year, and it is expected to double in the next two decades [6,7]. Therefore, in order to boost prosperity while alleviating dependence on the extraction of primary materials, more attention is being paid to sustainable concrete, where it is urgent to change the linear production to circular economy models.

Several studies [8,9] have identified that at the end of concrete’s useful life, the CDW of concrete can be crushed into particles of adequate size to be used as recycled concrete aggregates (RCA) in a new mix of recycled concrete (RC). Although RCA have demonstrated potential for reuse, their performance has limited their replacement level below 30%, mainly due to the porous and cracked cement paste layer adhering to their surface [10,11]. Thus, RCA have inferior properties, which include higher water absorption, which is up to five times higher than that of natural aggregates (NA) [12,13]. Consequently, the mechanical and durability behavior of RC made using RCA are lower than those made using NA [14,15].

Based on the feasibility of the improvement treatments reported in the literature, the water absorption of RCA in the mixture design of an RC is determinant, since all mixes need to be corrected for moisture to achieve the design w/c ratio [16]. Therefore, the different treatments in the RCA reduce the percentage of water absorption, minimizing the error associated with moisture adjustment and, in turn, improving mechanical and durability performance. The use of treatments from chemical, thermal and mechanical processes has been reported to improve the quality of RCA, which can be classified into two main groups: the first group focuses on weakening the residual bonded cement paste in RCA by heating/grinding [17,18,19], crushing with a jaw or ball mill [20,21,22], cleaning/scrubbing [23,24] or even soaking (acid) [25,26,27,28]; the second group focuses on strengthening the microstructure and surface of the RCA by using CO_2_ curing [29,30,31,32,33,34], biodeposition [35,36,37], coating (silica fume, fly ash, blast furnace slang) [38,39,40,41,42] and soaking (acid solution) [43,44,45].

According to a literature review, the authors Diana et al. [46] found that in contrast to weakening treatments, with strengthening treatments, an efficiency in relation to water absorption reduction has been reported for fine RCA that is six times higher and is approximately double for coarse RCA, which is mainly achieved from a balance between the type of treatment and the size of the RCA, since both the pore structure and the surface of the RCA are modified, which is noticed in the enhancement of the reactions of the hydration products and in the creation of new products that allow the pore structure to be refined. In contrast, improvement in the quality of coarse RCA has been more frequent with treatments to weaken the cement paste adhering to the RCA surface, mainly because the increase in particle size facilitates the removal of cement paste. Wang et al. studied the use of a diammonium hydrogen phosphate (DAP) solution, which is used commercially as an affordable soil fertilizer. By finding efficiency in terms of the microstructural densification of concretes made from treated coarse RCA due to the reaction between the DAP and the free calcium-rich hydration products (portlandite) on the surface of the RCA, one can produce hydroxyapatite (HAP) complexes, which are responsible for achieving a crack sealing and pore refinement effect [47]. 

The role of carbonation as a factor contributing to the degradation of reinforced concrete by depassivation occurs due to the decrease in the pH value [48], which is the main reason for damage to the protective layer of steel surfaces and the degradation of silicate in the paste. That is, according to the literature, calcium silicate hydrated (C-S-H) could be destroyed by the reaction with CO_2_ and water [49]. In contrast, carbonation could improve the surface conditions of the RCA, since the CO_2_ that enters by diffusion reacts with the available portlandite forming calcium carbonates, and consequently, a pore-sealing effect is achieved at the surface level [50]. In the case of RC, the coefficient of accelerated carbonation (k_ac_) has been studied, and the results suggest that k_ac_ mainly depends on the compressive strength of concrete regardless of the content, type and size of RCA, and also the w/c ratio of mixes [49]. 

This research aims to evaluate the effect of DAP treatment on fine RCA (RFA) and its relationship with the durability behavior of mortars exposed to aggressive agents such as CO_2_, taking into account that by improving the quality in terms of RFA pore refinement, DAP treatment should improve mortar durability because the formation of hydroxyapatite (HAP), as the main constituent, promotes the nonreaction of CO_2_ in the system, and also its high mechanical and chemical stability favors the pH balance.

## 2. Experimental Procedure

### 2.1. Tests

Water absorptionWater absorption was determined by using the innovative thermogravimetric balance halogen light (TBHL) technique, based on ASTM C138 for fine NA [51], which has demonstrated absorption values with less dispersion and optimizing resources such as energy, time and material. TBHL starts from the saturation state of the RFA to find the surface saturation dry state (SSD) via a thermogravimetric process using an XM 60-HR halogen light balance with the uniform and soft heating of the sample up to a maximum temperature of 85 °C. Real-time reproducible results are achieved by coupling to a ABSORPTION INNOVATION version 1.0 with a signal communication device (TTL to RS 234 level conversion) [51,52].Optical Microscopy (OM)Optical microscopy analysis was carried out in order to determine the morphological characteristics of the shape, texture and gradation of the RFA. Micrographs were taken with a Nikon stereoscope, reference Eclipse LV100, objectives 2× up to 11.5×.X-ray diffraction (XRD)A mineralogical composition via XRD analysis was carried out with the aim of identifying the phases formed from the DAP treatment in both the RFA and the mortars made from treated RFA. A PANalytical X “Pert PRO MPD reference instrument was used, in a range of 2θ between 15–36°, with a step of 0.02°. A copper anode with Kα_1_ = 1.5406 Å was used. Peak positions and relative intensities were compared with the X’Pert Hihg Score Plus license software database.Thermogravimetric analysis (TGA)TGA was used to characterize the thermal evolution of the dry precipitates identified with XRD in the case of RFA. The analysis was performed under a nitrogen atmosphere and continuous heating with a temperature range between ambient 30–380 °C at 10 °C/min and 380–800 °C at 5 °C/min to study the mass loss behavior of the precipitate. Peak positions and mass loss quantification was processed with the license-free software tool TRIOS V5 1.1.Scanning Electron Microscopy (SEM)The morphological characteristics and microstructure of the RFA powder products and mortars were characterized using a JEOL JSM 6490 LV high-vacuum scanning electron microscope for high-resolution images, using a secondary electron detector. The samples were fixed on a graphite tape and coated with gold (Au) by using the DENTON VACUUM Desk IV. An elemental analysis was performed using an X-ray Microprobe-EDX by INCA PentaFETx3 Oxford Instruments.Flow and pH of the mortar mixThe behavior of the fresh mortar was evaluated using the Standard Test Method for Flow of Hydraulic Cement Mortar (ASTM C1437) and the pH value with a peachimeter to validate if the DAP has an effect on the pH value of the mix.Compressive strengthThe mechanical behavior of the mortars manufactured with the treated RFA was determined by using a Controls press, Model CT-0151/E. A measuring range between 0–150 KN and a loading rate of 1250 N/s was used.Accelerated carbonationThe accelerated carbonation process was performed by using a continuous flow of CO_2_ (4%) and the gas flow rate (a mixture of pure CO_2_ and air) was controlled from 1.0 to 10 L/min. The exposure time was 28 days with a relative humidity and temperature of 60 ± 5% and 23 ± 5 °C, respectively.Carbonation depthAccording to RILEM guidelines, a pH change indicator was used to determine the depth of carbonation in the mortar by using a small amount of phenolphthalein at a 1% concentration, which was applied to the inside of the faces after cutting cubes perpendicular to the poured face [53]. Additionally, an image analysis was performed with the free ImageJ software version 1.44 to quantify the carbonation depth. A pH change indicator test was performed for comparative purposes, and to validate the results, XRD and SEM analysis were performed.

### 2.2. RFA Sourcing and Characterization

The RFA used were sourced from the crushing of young concrete waste from on-site quality test concrete cylinders. Table 1 presents the physical properties of the RFA and of an NA, which is used as a reference, and Figure 1 shows the transformation process of the waste concrete, in which a two-stage crushing system was used to obtain a particle size distribution according to the inferior curve (IL) of ASTM C33.

The morphology of the RFA is shown in Figure 2, based on the optical microscopy test, where an irregular and angular shape is evident and a porous cement paste adhered to the RFA surface, which may explain the higher water absorption rate of the RFA.

Figure 3 shows the identified mineralogical phases, which are partly typical for stone materials with a high quartz composition. Additionally, the presence of calcium hydroxide (Ca(OH)_2_) is associated with the presence of cement paste adhering to the surface of the RFA, with its main peaks on the 2θ axis (18° and 34°), as well as the presence of calcium carbonate (CaCO_3_), typical of the reaction between Ca(OH)_2_ and CO_2_ in the environment. These phases are associated with a high degree of crystallinity in the microstructure of the RFA.

### 2.3. DAP Treatment 

The DAP treatment was performed by soaking the RFA in a DAP solution based on a 3^2^ × 2 factorial experimental with 3 replicates, where three different concentrations (0.5 mol/L, 1 mol/L and 2 mol/L) of the DAP solution, three immersion times (1, 7 and 14 days) and two temperatures (20 and 40 °C) were used. Figure 4 presents a schematic illustration showing the steps followed in the DAP treatment, including the preparation of the DAP solution and the treatment conditions to achieve water uptake as a response variable.

### 2.4. Mortar Mix Design

The mortar mix design used structural Portland cement; the cement–sand (RFA) ratio was 1:2 by mass, and two w/c ratios were evaluated (0.45 and 0.50). All the mixtures were corrected for moisture, considering the water absorption value, from which the amount of additional water required by each mixture was known (Table 2). The mix proportions of the mortars made with treated RFA and the production process are shown in Table 3 and Figure 5, respectively.

## 3. Results and Discussions 

### 3.1. RFA Improvement with DAP Treatment

Table 4 presents the overall results of water absorption. In all the evaluated conditions of the DAP treatment, the treated RFA showed a lower water absorption compared to the untreated RFA, for which the water absorption value was 4.78%.

Figure 6 shows that the DAP treatment improved the quality of the RFA, with a maximum decrease in water absorption of 32.1% in the 0.5 mol/L–7 days–20 °C condition, and 28.9% in the 0.5 mol/L–7 days–40 °C condition. This reduction was induced by the reaction between the RFA and the DAP in the solution, which produced minerals that filled some pores and cracks of the RFA [47]. Although, these minerals were produced from the first day of immersion, demonstrating that most of the reduction in the water absorption was achieved at 7 days of immersion. Further reduction in water uptake can be achieved using a longer immersion time, but at a much slower rate due to the possible absence of CH to react with DAP. Even the increase in water absorption at 14 days is explained by the fact that HAP formation depends on the availability of CH to react, whereby the higher the concentration of DAP, the higher the demand for CH, so the absence of CH does not allow for HAP formation.

Figure 6 also shows that the water absorption of the RFA was maximally reduced at a concentration of 0.5 mol/L. When the DAP concentration was doubled or tripled, the water absorption was reduced to a lesser extent. These results are explained by two reasons: (i) the formation of new minerals that fill the RFA pores depends on the availability of Ca(OH)_2_ on the surface of the RFA to react with DAP, so in this sense, the higher the concentration, the higher the demand for Ca(OH)_2_, which is assumed to be insufficient; (ii) although the viscosity of the DAP solutions did not change much as a function of concentration, a 20% higher viscosity for the higher concentration of 2 mol/L may have limited the kinetic reaction due to greater opposition to entering inside of the RFA pores.

The pH values of all DAP solutions increased with immersion time, as shown in Figure 7. A rapid increase in the pH value was observed on the first day of immersion due to the dissolution of Ca(OH)_2_ from the RFA, which was much faster than the reaction rate between Ca(OH)_2_ and DAP. Afterwards, the increase in the pH value slowed down to a much slower rate because there was much less Ca(OH)_2_ from the RFA available. 

The higher the DAP concentration, the lower the pH value because more DAP was available to buffer the increase in the pH value induced by the dissolution of Ca(OH)_2_. In this sense, taking into account that the pH of the system was given by the Ca(OH)_2_ content, and that pH values between 10 and 11 enhance the formation of stable calcium phosphates as HAP, the treatment condition that strongly promoted the formation of HAP, and thus the improvement of RFA, was 0.5 mol/L–7d–20 °C.

On the other hand, the XRD patterns of the untreated RFA and RFA treated with 0.5 mol/L, 1.0 mol/L and 2 mol/L at 7 days and 20 °C are shown in Figure 8. For 0.5 mol/L–7d–20 °C, a decrease in the main peaks associated with Ca(OH)_2_ was identified at 18° and 34° of the 2θ axis, suggesting that it was consumed by the reaction with DAP, producing new minerals identified as calcium phosphates. In addition, a slight shift towards lower values was identified in the halo between 31° and 33° of the 2θ axis, which is decisive, considering that within this range, especially at 33°, the main peak associated with HAP was expected.

The reaction between Ca(OH)_2_ and DAP produced new mineral precipitates identified in the treated RFA as calcium phosphates (Ca_3_(PO_4_)_2_): dicalcium phosphate Ca(HPO_4_)2H_2_O, octacalcium phosphate CaH_2_(PO_4_)OH and Ca_3_(PO_4_)_2_, which can further react with OH^-^ and Ca^2+^ in the solution to produce HAP, (Ca_10_(PO_4_)_6_(OH)_2_) [47]. However, for concentrations of 1 mol/L and 2 mol/L, no HAP formation was identified, which could be explained by the low amount of Ca(OH)_2_ to react with HAP at high concentrations. Additionally, unlike HAP, the formation of dicalcium phosphate Ca(HPO_4_)2H_2_O and octacalcium phosphate Ca_8_H_2_(PO_4_)OH occurred in environments with low pH values and was therefore less stable [54]. These results explain the fact that the lower the DAP concentration, the greater the decrease in the adsorption value of the RFA, because the formation of a stable phosphate such as HAP allows for the surface strengthening and pore filling of the RFA, as is evidenced in Figure 8 for 0.5 mol/L.

The thermal analyses presented in Figure 9 were performed using untreated and treated RFA samples at 0.5 mol/L–7d–20 °C, with the aim of quantifying the amount of Ca(OH)_2_ that reacted with DAP to form the HAP. Four significant peaks were identified: The first one was assigned to free waters, and the second one to an unreacted amount of phosphate (DAP) remanence [55]. At higher temperatures, the third peak is associated with the dihydroxylation of Ca(OH)_2_, and the fourth with the decarbonation of CaCO_3_. These decompositions and the successive reactions could be associated with HAP formation, which was confirmed by the XRD results above. 

The CH (Ca(OH)_2_) weight losses for the untreated and treated RFA were 7.56% and 6.13%, respectively. Thus, it was found that the reaction efficiency of DAP with CH (hydration product of the cement paste adhered to the RFA surface) was about 19% (Figure 9).

The microstructure of the untreated and DAP-treated RFA is presented in Figure 10, for which a SEM/EDS analysis was performed. It can be observed in Figure 10a that regarding the untreated RFA, the surface of the RFA was highly porous and less homogeneous, unlike the treated RFA shown in Figure 10b: 0.5 mol/L–7d–20 °C and Figure 10c: 2.0 mol/L–7d–20 °C, where a denser and more homogeneous microstructure is shown. This was caused by the fact that some dissolvable particles on the surface of the RFA reacted with the DAP solution to produce new precipitates, which filled the pores and cracks of the RFA.

Figure 10b shows that the surface of the RFA was covered by the precipitate after being immersed in a 0.5 mol/L DAP solution for seven days, where the precipitate showed a spherical shape associated with a typical HAP morphology. On the other hand, no spherical precipitated particles were produced with the higher concentration of DAP, as shown in Figure 10c; however, there was a large presence of hexagonal particles, associated with unreacted CH, as well as a small number of precipitated particles with a slat-like structure (type fiber), which is associated with calcium phosphate crystals that can form in different habitats [54]. This confirms the finding of the XRD analysis shown in Figure 8.

EDX spectrums showed that in contrast to the spectrum in Figure 10b,c, Figure 10a shows only the presence of calcium (Ca) because there was no DAP treatment. When there was treatment, the presence of phosphate (P) from the DAP was noticeable; however, it is notorious that the higher the concentration of DAP, the higher the presence of P, which is associated with the impossibility of reacting due to the absence of Ca.

### 3.2. Durability Behavior of Mortars with Treated RFA

#### 3.2.1. Flow, pH and Compressive Strength of Mortars

Figure 11 shows the flow test results of the mixtures made from a partial and total replacement level of treated RFA (0%, 30%, 60% and 100%) and two w/c ratios (0.45 and 0.50). It can be observed that the lower the w/c ratio and higher the treated RFA substitution, the lower the flowability. The relationship between the w/c ratio and fluidity is associated with the demand for H_2_O for lower w/c ratios (see Table 3). On the other hand, the results indicate that there was a significant difference in flowability between the mixtures, since the higher the level of substitution of the treated RFA, the lower the flowability. The result is contradictory to that identified by Rolands et al., in which it is mentioned that the higher the specific surface area of particle, the higher the resistance to flow [56]. The tendency to decrease the flowability of the mixtures requires further study, since a high cohesion was experimentally found in the mixtures with higher proportions of treated RAP, so it is required to identify how the excess of phosphate ions on the surface of the treated RAP can interact rheologically with the Portland cement paste to define which phenomenon governs the flow or flowability of the mixtures.

The pH values of the mortar mixes presented in Table 5 allow us to identify that only a 5% change occurred in the mixtures with 100% treated RFA compared to 0% treated RFA. Regarding the possible residual ammonium in the treated RFA, studies have indicated that it could reduce the pH value of the cement paste and hinder the hydration processes [57]; however, in this case, the pH change in the system was not significant.

Figure 12 shows the compressive strength results of mortars with treated RFA according to the mix design in Table 3. In addition, by way of comparison, the compressive strength of mortars with 100% natural aggregate with a w/c of 0.45 at 28 days of curing is shown, whose average value was 65.39 MPa (0.055 of standard deviation). The presence of the treated RFA significantly increased the compressive strength of the mortars, especially for replacement levels of 60% and 100%, where compared to the 0% replacement, the strength was increased by 27% and 35%, respectively. Even if the average compressive strength values of the mortar with treated RFA and with natural aggregates are compared, a similar behavior is evident, and even slightly higher values are observed with a replacement level of 100% treated RFA, indicating that a DAP treatment can produce high-quality RFA. In turn, the higher the w/c ratio, the lower the compressive strength. This behavior is associated with a higher porosity in the cementitious matrix.

The SEM images of the mortar (0.45 and 0.50 of w/c) samples made with the untreated and treated RFA are shown in Figure 13. As shown in Figure 13a,c, for mortars made with the untreated RFA (0% replacement of treated RFA) with a w/c of 0.45 and 0.50, there was a large and loose interfacial transition zone (ITZ, between old cement paste and new mortar paste). This zone is induced by the cement paste adhering to the RFA and is responsible for the low compressive strength of the mortar found previously. For a w/c of 0.50, a clean surface was observed in the ITZ of untreated mortar; however, when the RFA were treated, both CH and calcium phosphates were observed in the ITZ, and it was evident that this ITZ was considerably denser and less wide.

On the other hand, when the untreated RFA were not included, a homogeneous texture was observed, as shown in Figure 13a,c, unlike the mortar with treated RFA, which presented a predominantly irregular texture associated with deposits of calcium phosphates and HAP identified in the mineralogical analysis by XRD in Figure 8.

Figure 13b,d show how the ITZ can be improved by consolidating the cement paste bonded to the RFA surface by DAP treatment. The images show how the ITZ of the mortar with 100% treated RFA was reduced, mostly for mortar with a w/c of 0.45 in Figure 13b. This may lead to an increase in the mechanical properties (Figure 12) due to the densification of the ITZ, which prevents crack propagation under stress.

The water–cement ratio had a significant effect on the microstructure, because the higher the w/c, the higher the porosity, which could be related to water demanding mineralogical phases such as the precipitation of ettringite which requires up to 32 moles of water [58], whose crystalline structure can be in the form of a prism with a hexagonal or acicular (needle) cross-section, which can be found in Figure 13d.

Finally, after validating the increase in the compressive strength of the mortars by the addition of DAP-treated RFA, it is concluded that the densification of the ITZ between the RFA (bonded cement paste) and the new cement paste by DAP treatment promoted the mechanical performance of the mortars by preventing the propagation of cracks, and this was induced by the porosity of the RFA.

#### 3.2.2. Carbonation Phenomenon of Mortar

Figure 14 shows the effect of the carbonation phenomenon on mortars made with different levels of RFA substitution treated with DAP (0%, 30%, 60% and 100%). It was observed that there was a change in the pH of the mortars, indicating the presence of carbonation (nonpink area). Mortars with 0% DAP-treated RFA had greater and unequal carbonation fronts, and as the substitution of the untreated RFA with the treated RFA increased, a decrease in the carbonation front was observed. 

From Figure 14, it can be said that the maximum carbonation front in percentage area was obtained for the mortars made with untreated RFA (0% treated RFA) for both a w/c of 0.45 and 0.50. That is, when including 100% treated RFA, the carbonated area was only 21% and 34% for the mortars with 0.45 and 0.50 w/c, respectively. However, it should be noted that although this behavior was found both for mortars manufactured with a w/c of 0.45 and 0.50, the first ratio showed a lower carbonation front, where the carbonation area was lower by more than two times in all cases. This behavior is associated with the porosity of a cementitious matrix with a high w/c [7].

The carbonation depth of mortars manufactured with treated RFA is presented in Figure 15, where a decrease was observed as a function of the level of the treated RFA, showing that the higher the percentage of the treated RFA, the lower the carbonation depth, which means that the carbonation depth when substituting 100%, 60% and 30% of the treated RFA with DAP was, respectively, 90%, 57% and 38% lower than that of the control (0% of treated RFA) for a w/c ratio of 0.45, and 65%, 53% and 30% for a W/C of 0.50. These results are in agreement with several studies [59,60,61], which concluded that the carbonation depth increases with an increasing amount of untreated RFA, where the quality of the RFA has been shown to have a great influence not only on the compressive strength but also on the carbonation resistance.

The XRD analysis carried out on the samples of carbonated and noncarbonated mortar with 100% treated RFA and with a w/c of 0.45 is presented in Figure 16, the result of which validated that the carbonation found through the change in pH was due to the presence of calcium carbonates with great intensity in the characteristic peaks 29° and 39° of the 2θ axis, which was a result of the reaction of CO_2_ that entered through the pores from the air and formed carbonic acid when reacting with calcium hydroxide [62]. 

The CO_2_ diffusion process was modeled using the solved equation for Fick’s first law. The model for predicting the carbonation depth over time was based on the diffusion law and considered the carbonation rate to be proportional to the square root of the CO_2_ exposure time; see equation 1, where *x* is the carbonation depth obtained for each condition in Figure 15, *k_ac_* is the carbonation coefficient obtained from *x* assuming a constant CO_2_ concentration and quantity, and *t* is the 28-day CO_2_ exposure time of the mortars in the accelerated carbonation chamber.
(1)$$x=k_{ac}\sqrt{t}\;$$

Based on the Reglamento Colombiano de Construcción Sismo-Resistente NSR-10-title C, Table 6 shows that taking into account a projected durability before exceeding a depth of the reinforcing steel cover equivalent to 40 mm, the carbonation phenomenon had no significant effect on the durability of the mortars made with DAP-treated RFA with w/c ratios of 0.45 and 0.50.

The results presented in Table 6 could be associated not only with the microstructural densification, achieved by the use of the DAP treatment in the RFA where the formation of calcium phosphates such as hydroxyapatite (HAP) as the main constituent (Figure 10) have been largely decisive in preventing the access of CO_2_ into the system, but also with the pH value regulation in the system due to the chemical stability of HAP, staying above 11, as evidenced in the fresh state of the mortar mixes (Table 5) as well as in the pH indicator measurements of the mortars in a hardened state (Figure 14). In addition, the remaining or residual DAP in the system could also inhibit the corrosion of the embedded steel. This was because the presence of ${\mathrm{PO}}_{4}^{3^{-}}$ could form an iron phosphate precipitate (FePO_4_·2H_2_O) on the surface of the steel, which acts as a physical barrier to protect from corrosion [63,64]. 

#### 3.2.3. Correlation between Compressive Strength and Carbonation Phenomenon

The results of the correlation of the 28-day compressive strength of the carbonated (f’c_x_) and noncarbonated (f’c_y_) mortars with different levels of treated RFA at 0.5 mol/L–7d–20 °C are presented in Figure 17. The comparison of the compressive strength means, whose premise assumes that the populations are normally distributed and independent, allows us to know that for a w/c of 0.45 and 0.50 and all treated RFA substitution levels, the statistical test (t_o_) versus the critical value (t_α/2_) indicated that there was no significant difference between the compressive strength means, meaning that under the evaluated conditions, the carbonation phenomenon did not have a negative effect on the durability in terms of compressive strength.

Figure 18 shows a strong correlation between the compressive strength and the coefficient of accelerated carbonation (k_ac_) of the mortars as a function of the substitution level of the treated RFA at 0.5 mol/L–7d–20 °C for a w/c of 0.45 and 0.50.

The carbonation rate (k_ac_) presented an inversely proportional relationship with the compressive strength for both w/c ratios, showing that higher percentages of the treated RFA favored the increase in the compressive strength and the decrease in the carbonation rate. This behavior is associated with the higher compactness and lower permeability of the cement matrix as one of the consequences of the microstructural densification obtained by the use of the DAP treatment in the RFA. In other words, by achieving a decrease in the water absorption of the RFA due to the filling of the pores, the lower moisture demand (moisture correction, see Table 4) led to more compact matrices with fewer capillary pores, thus presenting a lower diffusion of CO_2_. The results are in agreement with Brito et al. [61], who found that compressive strength is an indirect measure of compactness and a good indicator of the permeability and durability of concretes.

Figure 19 shows the effect of the carbonation phenomenon on the interfacial transition zone (ITZ) of the mortars made with 0% and 100 % replacement of treated RFA: uncarbonated mortars (Figure 19a,c) and carbonated mortars (Figure 19b,d). It was found that the mortars with untreated RFA in Figure 19a,b presented a large and nonhomogeneous ITZ, and this was especially true for the mortars exposed to carbonation, whose width extended and promoted the carbonation depth presented in Figure 14. This larger ITZ also caused more damage to the mortar matrix, and over time this damage was not primarily confined to the ITZ but started to spread to the surrounding mortar elements. 

On the other hand, when a 100% replacement of RFA treated with DAP was used, a predominantly dense microstructure with a less evident ITZ was noted, even when the mortar was exposed to accelerated carbonation, as shown in Figure 19d. This result could be explained by the filling of the pores and cracks of the treated RFA, which favored densification in the cementitious matrix. 

When comparing the ITZ results of the carbonated mortars with treated and untreated RFA (0% and 100%), the average ITZ width increased five times, see Figure 19b,d. This explains the fact that the DAP treatment did have a positive effect on the durability of the mortar, because by improving the quality of the RFA, the system in a new cementitious matrix became more densified and therefore the diffusion of external agents such as CO_2_ became more difficult.

## 4. Conclusions

The study of DAP treatment on RFA, evaluated by the concentration of the DAP solution, immersion time of RFA and the treatment temperature, allowed us to conclude that:

The DAP treatment presented a positive effect on the refinement of the pores as a consequence of the formation of hydroxyapatite-type calcium phosphates produced by the reaction between the portlandite on the surface of the RFA and the DAP, achieving a reduction in the water absorption of the RFA of up to 33%.

The presence of the treated RFA significantly increased the compressive strength of the mortars, especially for the replacement levels of 60% and 100%, where compared to 0% replacement, the strength was increased by 27% and 35%, respectively. Even when the average compressive strength values of the mortar with treated RFA and with natural aggregates were compared, a similar behavior was obtained.

The durability of mortars manufactured with DAP-treated RFA was not affected by the carbonation phenomenon, as shown by the compressive strength values, for which no decrease was observed, and in the carbonation behavior, where the carbonation depth decreased by up to 90% and 63% for mortars with a w/c of 0.45 and 0.50, respectively.

The carbonation rate (k_ca_) presented an inversely proportional relationship with the compressive strength for both w/c ratios of 0.45 and 0.50, showing that higher percentages of treated RFA in the mortar favored the increase in compressive strength and the decrease in carbonation rate, where the closest relationship was found for a substitution of 100% treated RFA, which was 90% higher than the mortar with 0% treated RFA; the behavior associated with the microstructural densification was obtained from the use of the DAP treatment.

Considering the potential use of treated RCA, this study concludes that it is necessary to advance in the regulation of the incorporation of treated fine and coarse RCA; additionally, it is necessary to show companies the need to optimize natural resources, taking advantage of the large amount of test cylinders that are produced in the works only for the quality control of concrete. In Colombia, for example, according to the Reglamento Colombiano de Construcción Sismo Resistente NSR-10, title C, the construction of structural elements requires the evaluation of three cylinders per 40 m^3^ and 200 m^2^ and per type of mix.

## Figures and Tables

**Figure 1 materials-16-00980-f001:**
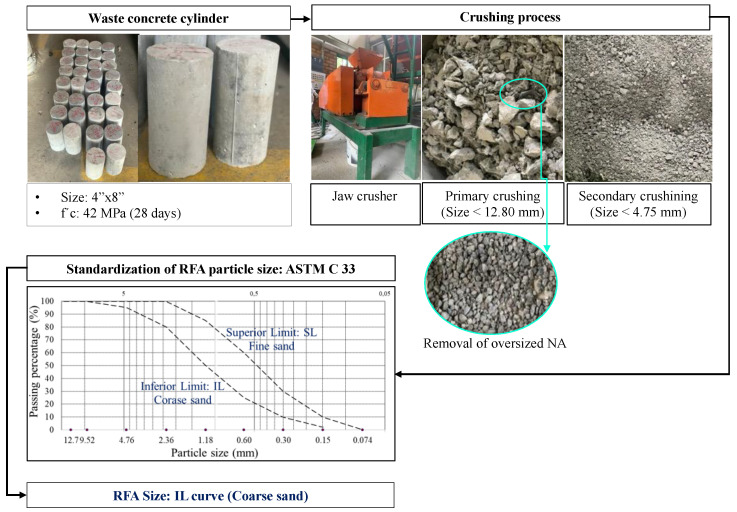
Schematic illustration of transformation process of concrete waste.

**Figure 2 materials-16-00980-f002:**
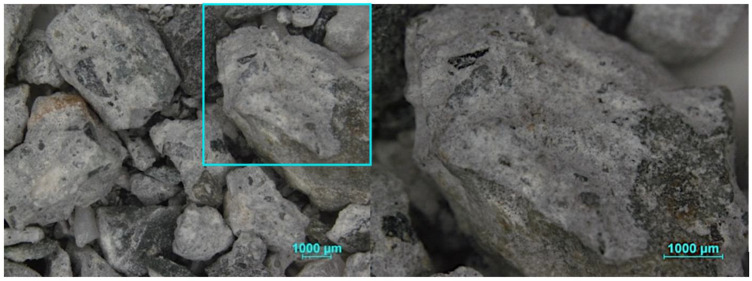
Micrographs of RFA.

**Figure 3 materials-16-00980-f003:**
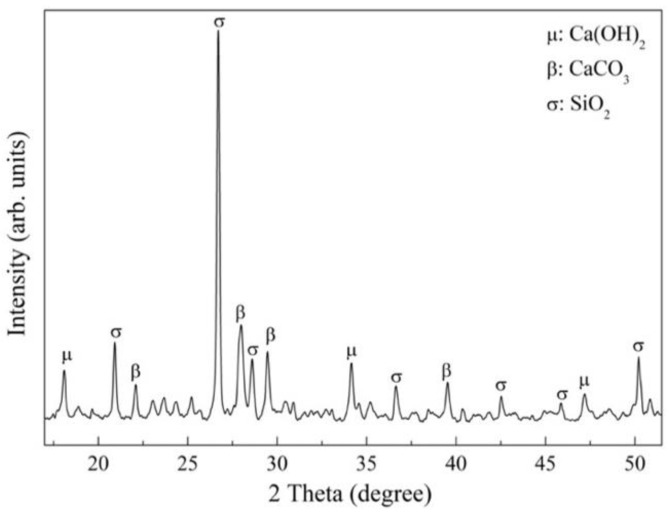
Mineralogical composition of RFA.

**Figure 4 materials-16-00980-f004:**
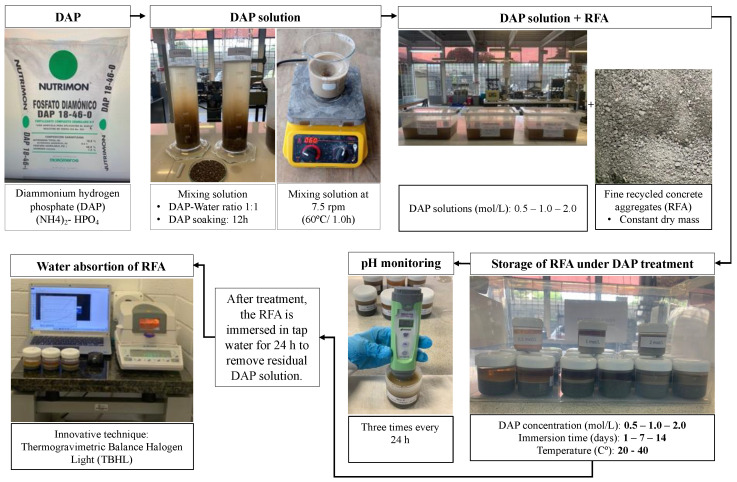
DAP treatment process.

**Figure 5 materials-16-00980-f005:**
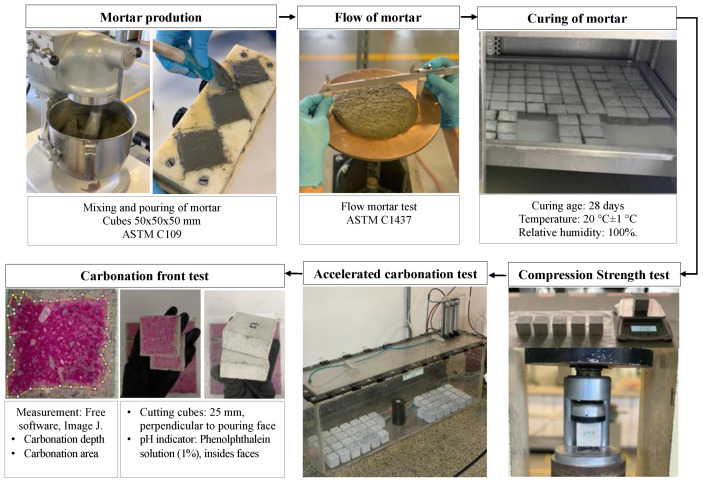
Mortar production and evaluation.

**Figure 6 materials-16-00980-f006:**
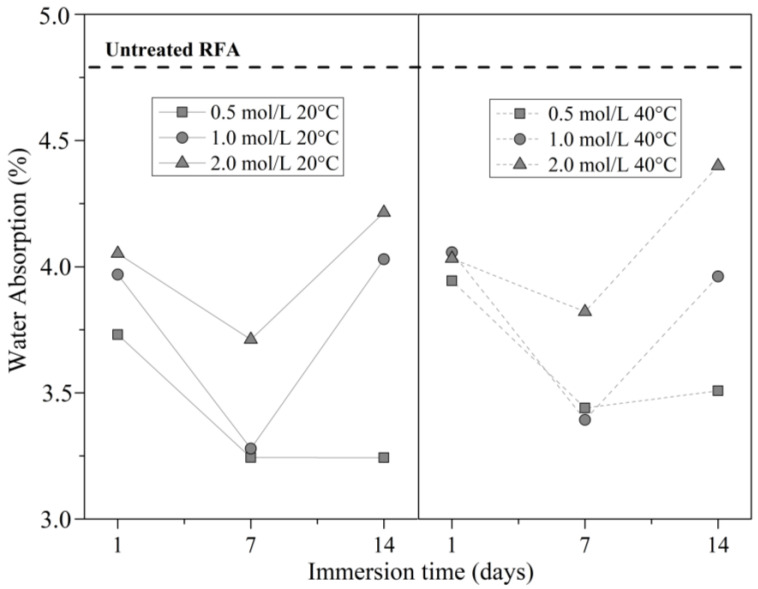
Water absorption (%) of the RFA treated with DAP solution.

**Figure 7 materials-16-00980-f007:**
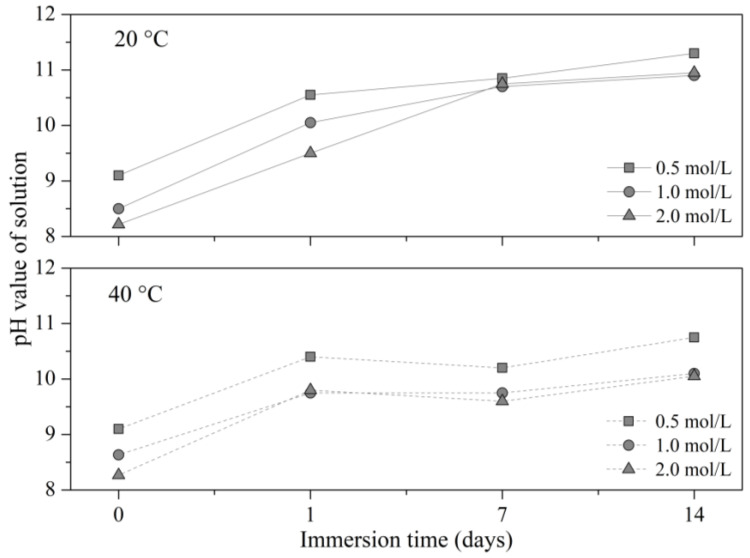
pH values of the DAP solutions varying with the immersion time and temperature.

**Figure 8 materials-16-00980-f008:**
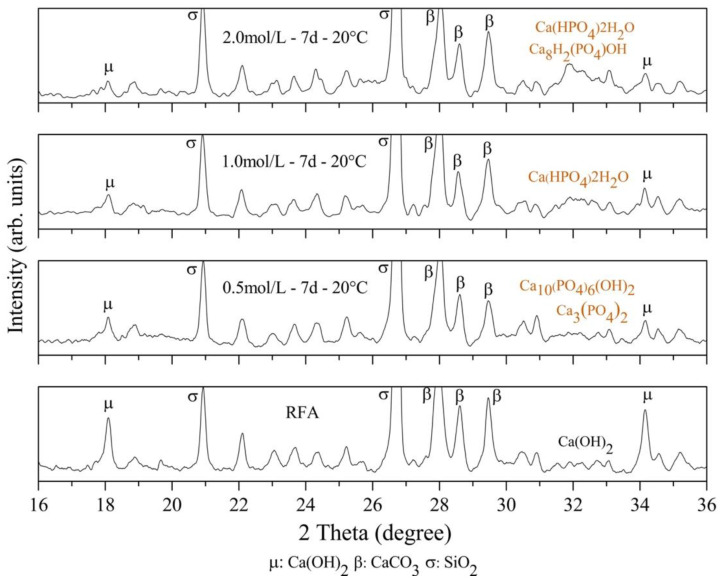
XRD patterns of the RFA treated with 0.5, 1.0 and 2.0 mol/L DAP solution.

**Figure 9 materials-16-00980-f009:**
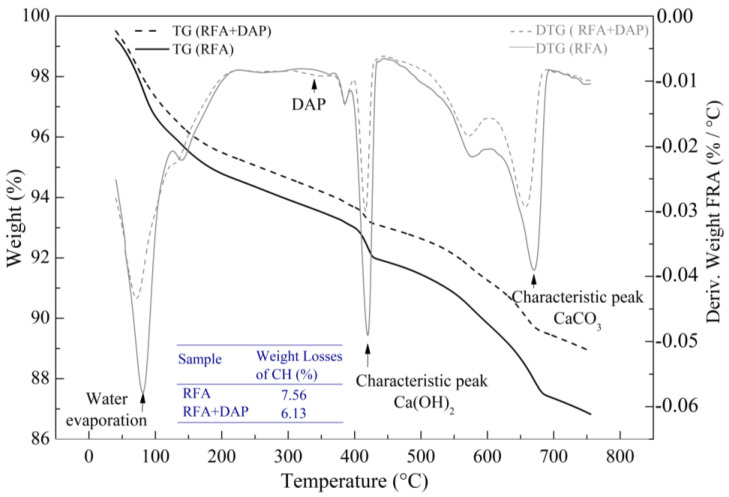
TGA results of RFA and RFA+DAP samples at 0.5 mol/L–7d–20 °C.

**Figure 10 materials-16-00980-f010:**
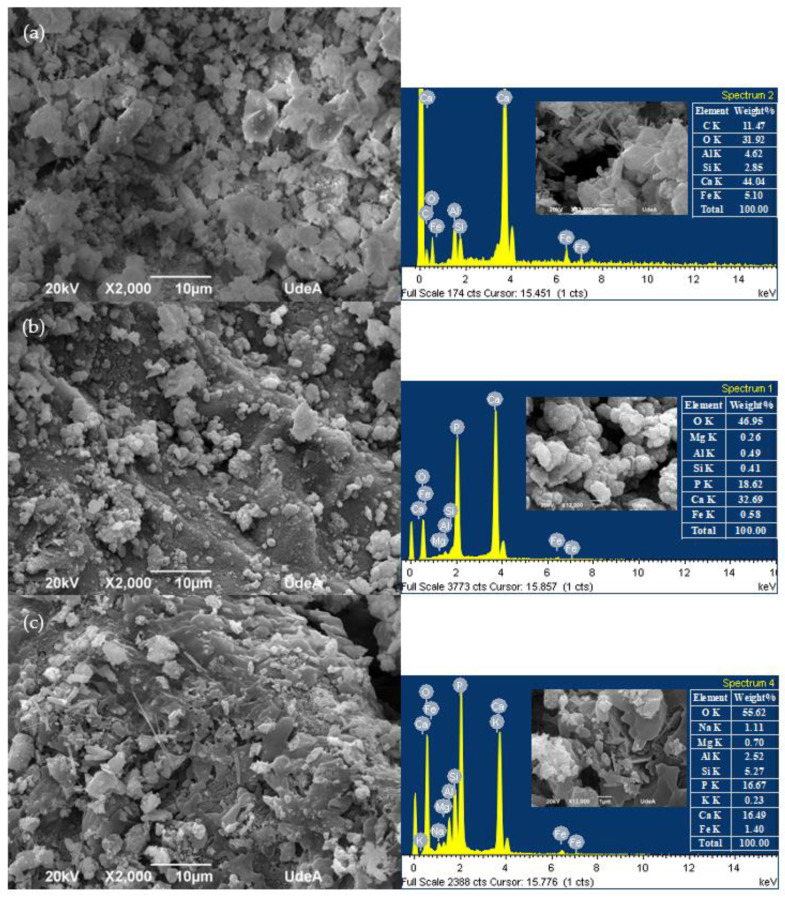
SEM images of the untreated and treated RFA with DAP solution: (**a**) untreated RFA; (**b**) 0.5 mol/L–7d–20 °C; (**c**) 2.0 mol/L–7d–20 °C.

**Figure 11 materials-16-00980-f011:**
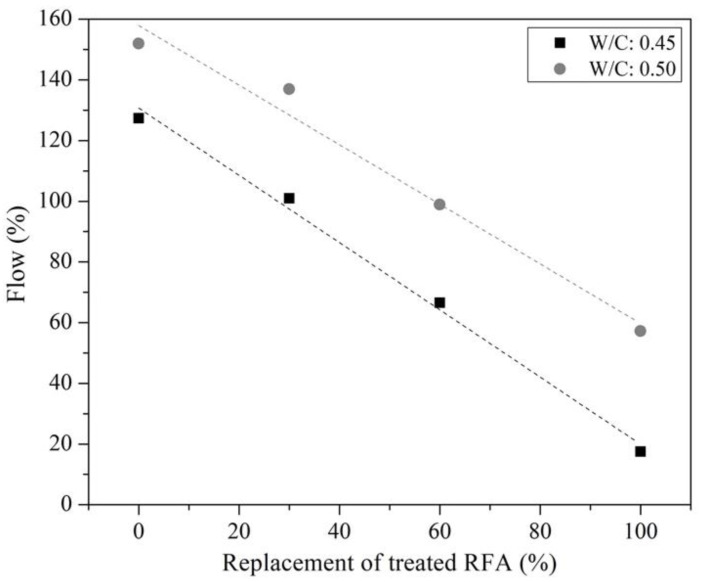
Flow of mortar mixes with different replacement of treated RFA at 0.5 mol/L–7d–20 °C.

**Figure 12 materials-16-00980-f012:**
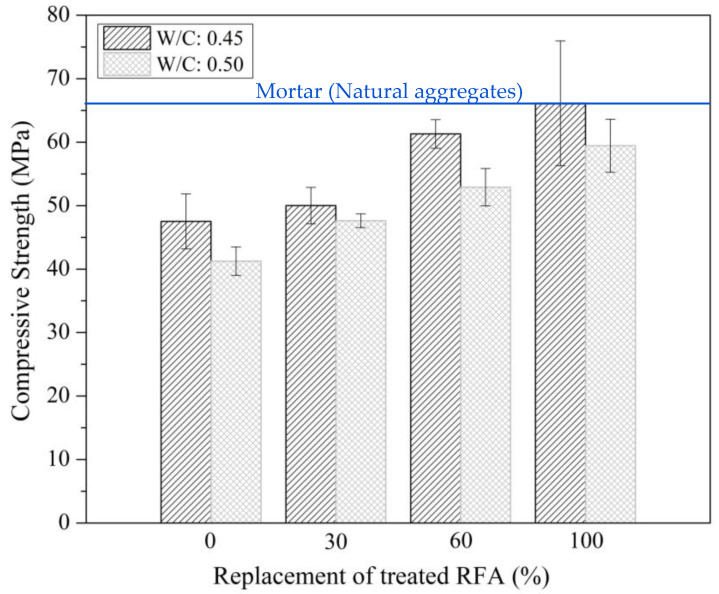
28-day compressive strength for mortars with different replacement of treated RFA at 0.5 mol/L–7d–20 °C and natural aggregates.

**Figure 13 materials-16-00980-f013:**
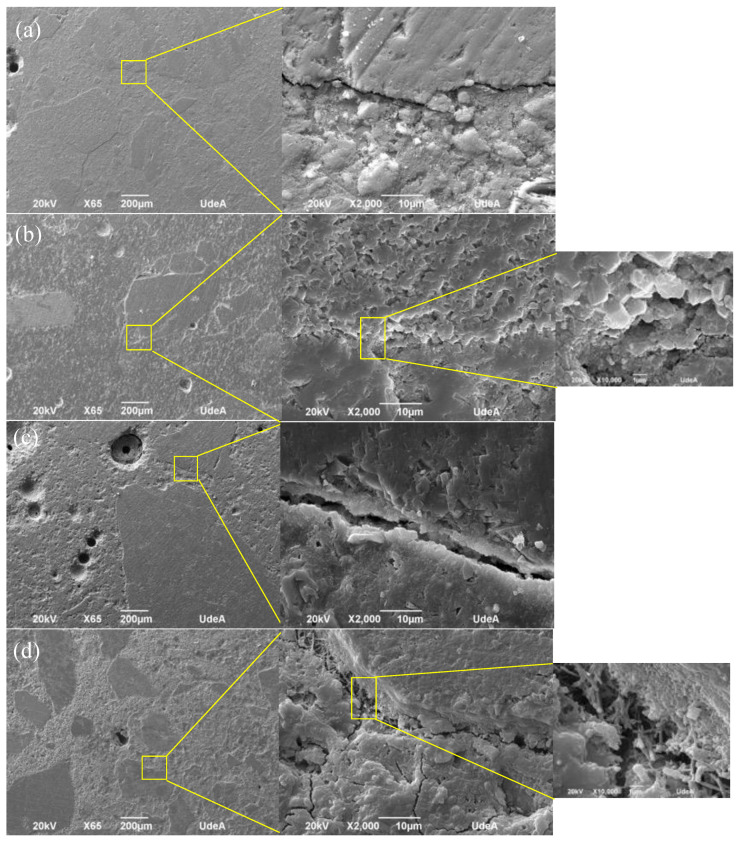
Microstructure of mortars made with RFA: (**a**) mortar made with 0% treated RFA, w/c: 0.45; (**b**) mortar made with 100% treated RFA (0.5 mol/L–7d–20 °C), w/c: 0.45; (**c**) mortar made with 0% treated RFA, w/c: 0.50; (**d**) mortar made with 100% treated RFA (0.5 mol/L–7d–20 °C), w/c: 0.50.

**Figure 14 materials-16-00980-f014:**
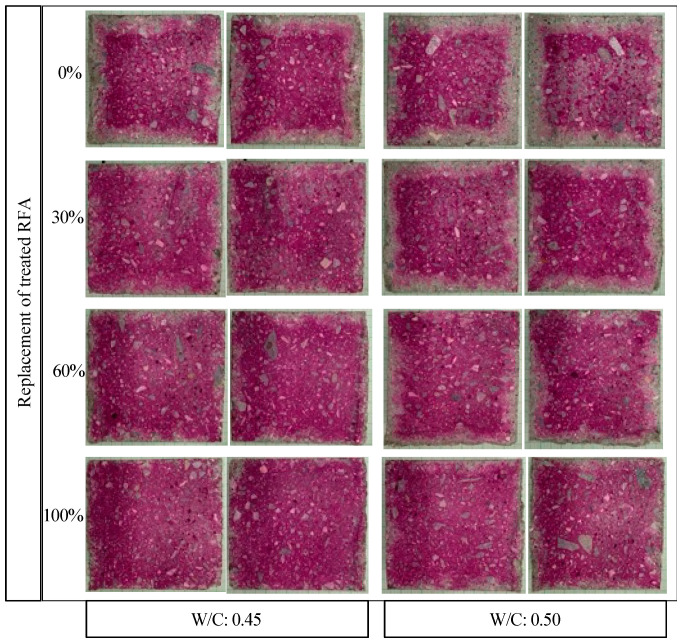
Carbonation front of mortar made with different replacement of treated RFA at 0.5 mol/L–7d–20 °C.

**Figure 15 materials-16-00980-f015:**
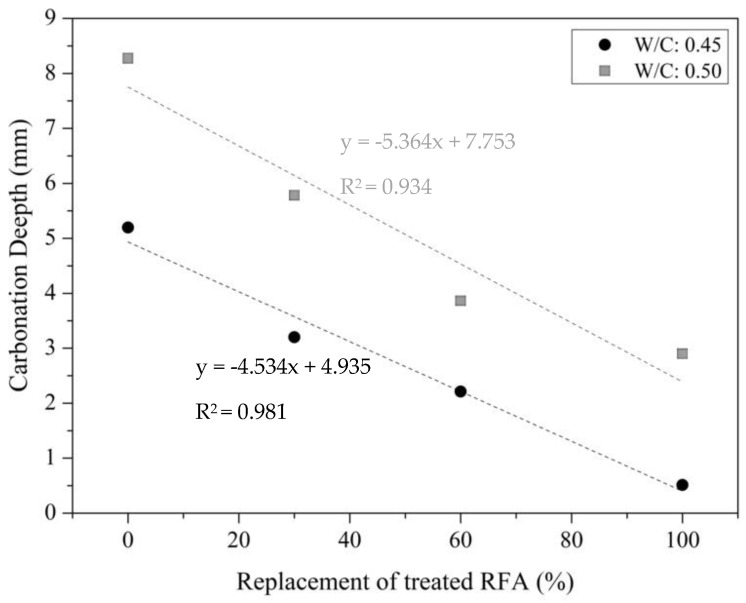
Carbonation depth of mortar made with different replacement of treated RFA at 0.5 mol/L–7d–20 °C.

**Figure 16 materials-16-00980-f016:**
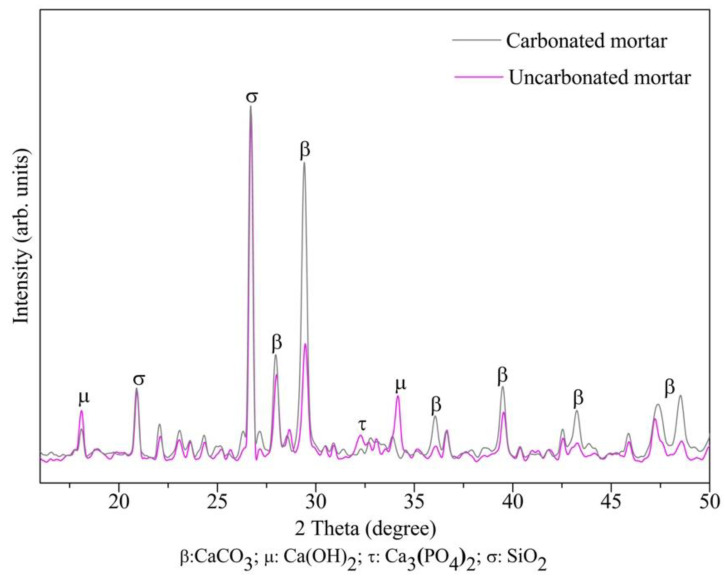
XRD patterns of carbonated (0% treated RFA) and uncarbonated (100% treated RFA) mortar.

**Figure 17 materials-16-00980-f017:**
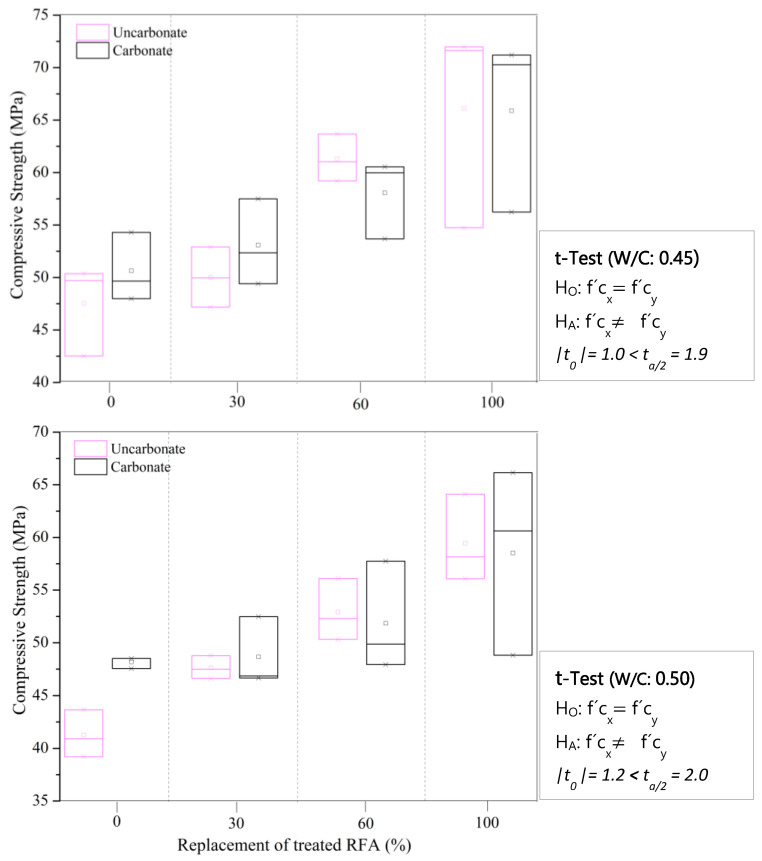
Correlation of 28-day compressive strength with carbonated and uncarbonated mortars with different replacement of treated RFA at 0.5 mol/L–7d–20 °C.

**Figure 18 materials-16-00980-f018:**
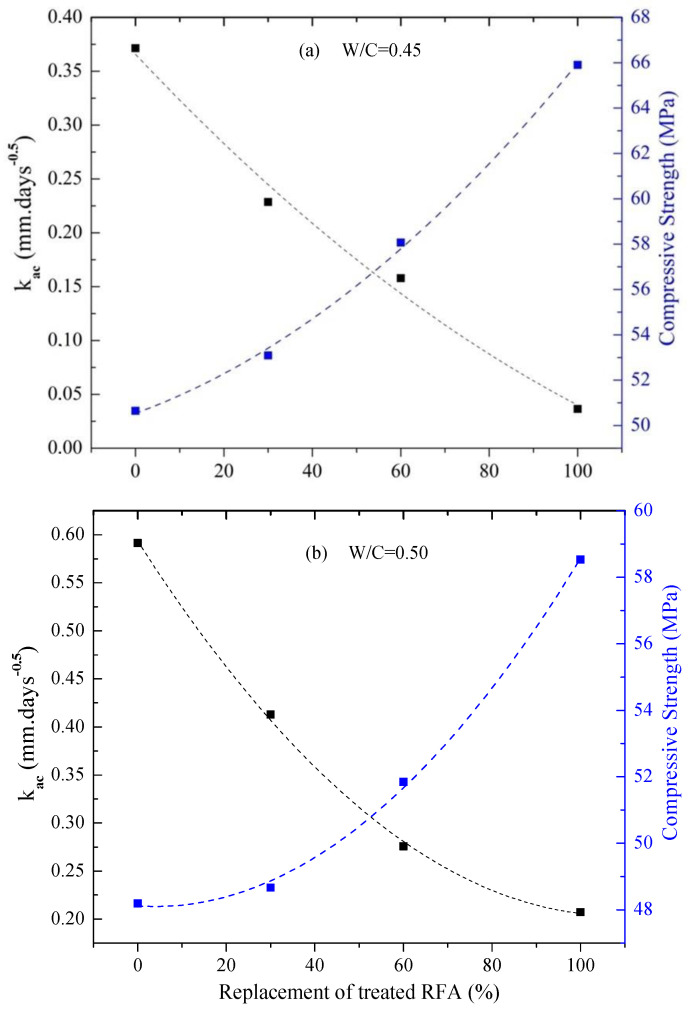
Correlation between k_ac_ and the 28-day compressive strength for mortar with different replacement of treated RFA. (**a**) W/C: 0.45; (**b**) W/C: 0.50.

**Figure 19 materials-16-00980-f019:**
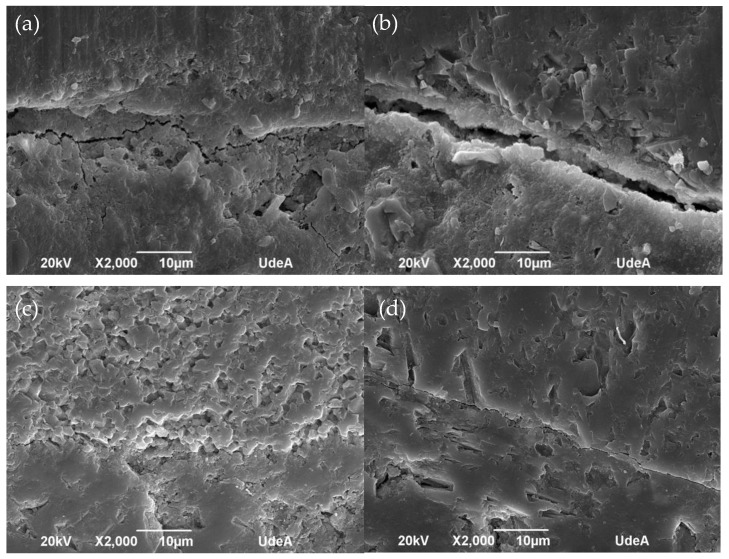
SEM observation of ITZ of mortar: (**a**) uncarbonated (0% treated RFA); (**b**) carbonated (0% treated RFA); (**c**) uncarbonated (100% treated RFA); (**d**) carbonated (100% treated RFA).

**Table 1 materials-16-00980-t001:** Physical proprieties of RFA.

Property	RFA	NA	Standard
Fineness	3.10	2.80	ASTM C 33
Bulk density (kg/m^3^)	1225	1615	ASTM C 128 and TBHL
Water absorption (%)	4.78	1.25	

**Table 2 materials-16-00980-t002:** Water absorption value for RFA.

Replacement of Treated RFA (%)		Water Absorption (TBHL Test)
0%	0%-Treated RFA + 100%-Untreated RFA	4.78
30%	30%-Treated RFA + 70%-Untreated RFA	4.32
60%	60%-Treated RFA + 40%-Untreated RFA	3.86
100%	100%-Treated RFA + 0%-Untreated RFA	3.24

**Table 3 materials-16-00980-t003:** Mix proportion of the mortar made with treated RFA.

Mortar Type	w/c	Mix Proportions					
		Cement (g)	Treated RFA (g)	Untreated RFA (g)	Water (g)	Total Water (g)	Additional Water
0%/0.45	0.45	600	0	1200	270	327.29	18%
0%/0.50	0.50		0	1200	300	357.29	16%
30%/0.45	0.45		360	840	270	321.78	16%
30%/0.50	0.50		360	840	300	351.78	15%
60%/0.45	0.45		720	480	270	316.27	15%
60%/0.50	0.50		720	480	300	346.27	13%
100%/0.45	0.45		1200	0	270	308.92	13%
100%/0.50	0.50		1200	0	300	338.92	11%

**Table 4 materials-16-00980-t004:** Water absorption (%) and its reduction in the treated RFA.

Immersion Time (Days)	Temperature (°C)	0.5 mol/L			1 mol/L			2 mol/L		
		Value	σ	Reduction	Value	σ	Reduction	Value	σ	Reduction
1	20	3.73	0.06	21.9%	3.97	0.17	16.9%	4.05	0.11	15.1%
7		3.24	0.15	32.1%	3.28	0.03	31.3%	3.71	0.06	22.3%
14		3.24	0.08	32.1%	4.03	0.04	15.6%	4.21	0.05	11.7%
1	40	3.94	0.05	17.4%	4.06	0.18	15.0%	4.03	0.12	15.5%
7		3.44	0.00	28.0%	3.39	0.00	28.9%	3.82	0.08	20.0%
14		3.51	0.02	26.5%	3.96	0.16	17.0%	4.40	0.11	7.9%

**Table 5 materials-16-00980-t005:** pH values of the mortar mixes with different replacement of treated RFA at 0.5 mol/L–7d–20 °C.

Replacement of Treated RFA	pH of Mix			
	w/c: 0.45		w/c: 0.50	
0%	12.00	0.23	12.10	0.05
30%	11.79	0.24	11.75	0.15
60%	11.38	0.04	11.50	0.16
100%	11.37	0.02	11.26	0.17

**Table 6 materials-16-00980-t006:** Durability of mortar made with different replacement of treated RFA at 0.5 mol/L–7d–20 °C.

Treated RFA (%)	t (Years)	
	W/C = 0.50	W/C = 0.45
0	13	32
30	26	84
60	58	176
100	102	438

## Data Availability

The data are available upon request from the corresponding author.

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
