# Peer review of "Carbonation Behavior of Mortar Made from Treated Recycled Aggregates: Influence of Diammonium Phosphate"

_materials, 2023, doi:10.3390/ma16030980_

Round 1

Reviewer 1 Report

The manuscript presents an interesting experimental work that investigates the effects of the diammonium hydrogen phosphate treatment on fine recycled aggregates (RFA) as well as on the behavior of mortars with (RFA) treated or not. The paper is well written and organized; the following are some points that I suggest addressing to improve the paper.

1)     I recommend elimination from line 19 to the beginning of line 21 “ The carbonation …non-carbonated”, since I think it does not present one of the most important conclusions of the paper.

2)     Add the reference number to Diana et al., line 65 in the Introduction.

3)     Rephrase the sentence of lines 80-82.

4)     Add to section 2.1 “Tests” the description of the flow tests on mortars, and the procedure for ph measurements for RFA and mortars, since the results of these tests are reported in the paper without the description of the experimental procedure.
Moreover, for sake of clarity, add that accelerated carbonation is made on mortars.

5)     It would be very useful to consider also a reference mix with only natural aggregates (NA), for both W/C equal to 0.45 and 0.50, and then compare the properties (flow, compressive strength, SEM images, XRD, carbonation depth) of all mortar types of Table 3 with also this reference mortar.

6)     Section 3.2.1 does not report only the results of compressive tests, but also flow and ph measurements. Change the title of section 3.2.1 or put these measurements in two additional sections.

7)     In figure 11, why does the flow decrease if the same effective water is used for all the mixes? If the additional water perfectly compensates for absorption, the flow should not change. The explanation of the obtained results - lines (303-306) - is unclear. 

8)     What do the compressive strength values in lines 319,3320 mean? Please rephrase the sentence and check the values.

9)     Figure 12 reports in blue a rectangular defining the mortar with natural aggregates. Was a reference mortar with only natural aggregates cast? If yes, report its precise value and the corresponding standard deviation.

10)  For a better understanding, I think that lines 385-395 and the corresponding Figure 15 should be moved after lines 395-405 and figure Figure 16.

11)  For the coefficient of accelerated carbonation (kac) assume the same format (not capital and small letter, i.e. in the text vs. Figure 18). Moreover, if it can be considered the same parameter of equation (1), use the same format also in this case.

12)  The compressive strength values in Figure 17 for uncarbonated samples are different from that in Figures 12 and 18. Please check the values or explain the difference.

13)  Rephrase the first sentence of the conclusion (lines 497-503), because it is too long.

14)  Correct some misprints:

-add the point at the end of the sentence, line 90;

- add the point to the total water amount (346.27 instead of 346.27) of mortar type 60%/0.50 in Table 3;

- use the same font as the rest of the text for lines 262-268;

- remove the capital letter of Results, line 464.

Author Response

Dear reviewer, thank you very much for your thoughtful review and suggestions, we have worked on each of them. All your recommendations have helped us to significantly improve our paper.

Reviewer 2 Report

This work deals with the evaluation of durability in mortars with recycled fine aggregates treated with diammonium hydrogen phosphate (DAP). Different mixtures with w/c ratios of 0.45 and 0.50 are evaluated, with DAP concentrations of 0.5, 1 and 2mol/L, with immersion times in DAP of 1, 7 and 14 days at 20 and 40ºC. Experimental tests of Water absorption, Optical microscopy (OM), x-ray diffraction (XRD), thermogravimetric analysis (TGA), scanning electron microscopy (SEM), compressive strength and accelerated carbonation were carried out. Expressions are proposed for the coefficient of accelerated carbonation and for the uniaxial compression strength as a function of the recycled fine aggregate percent treated with DAP.

Observations:

The manuscript is well written and organized. However, the English language should be checked, because there are some grammatical and writing errors, such as:

1- In line 65 instead of saying “Diana et al…” it is preferably to replace it by “the authors…” or to reformulate the sentence using passive voice.

2- In line 402, please, review the expression “are IN agreement ABOUT/ON…”

The bibliography is acceptable updated, but it is unnecessarily extensive, especially in the introduction lines 56 to 64. The reviewer advises only citing the most relevant works.

It is clear that different percentages of recycled fine aggregate are treated with DAP, but It is not clear if all fine aggregates are recycled, or some part corresponds to natural fine aggregates.

In lines 323 to 326 the authors mention that mortars with greater f’c have a higher quality. That concept should be reviewed. It is true that f'c is higher with a higher percentage of recycled fine aggregate (RFA) treated with DAP, but also a greater dispersion is observed in the results of f'c with 100% treated RFA, to what is this dispersion attributed?

The text in lines 401 to 405 is more conducive for the introduction, not for discussion of results.

In lines 467 to 471, is it possible to obtain an analytical expression of the coefficient of accelerated carbonation (Kac) as a function of the compressive strength, the w/c ratio and the percentage of aggregate treated with DAP?.

Author Response

(The authors gave the same response as above.)

Reviewer 3 Report

-We live now in a climate emergency so its most strange that the authors have not start the paper by mentioning exactly that. It seems that they are not aware about the words of a Professor of Physics at the University of Oxford authored a paper where one can read the following:

 “Let’s get this on the table right away, without mincing words. With regard to the climate crisis, yes, it’s time to panic”

Pierrehumbert, R., 2019. There is no Plan B for dealing with the climate crisis. Bulletin of the Atomic Scientists, pp.1-7.

So please start the introduction by draw a connection between environmental degradation and resource efficiency.

- The first phrase of the abstract must summarize the introduction.

- “ a foreseen increase in the demand for the next decades”

Comment: Provide data

- “with about 30 Gt/year of CDW, from which about 70% are CDW of concrete”

Comment: Please note that only rich countries have accurate statistics on CDW generation. Gide details about geographical location of that data

- “The use of diammonium hydrogen phosphate (DAP) solution, has demonstrated efficiency”

Comment: Mention the name of the author that first use this treatment. Also give more details about the solution. Is it expensive ? Is it toxic ?

- Why the use of halogen light thermogravimetric technique for determining water absorption ? What is the problem of using hydrostatic weighing ?Can this be a case of expensive redudant technology ?

- What was the standar that was followed for the accelerated carbonation test ?

- The authors used phenolphthalein for assessing carbonation but this is a toxic material that can cause genetic defects and even cancer. Why not use a different material or process ? Let´s not forget that some authors wrote that “The use of the phenolphthalein indicator underestimate real concrete carbonation by as much as 100%Torgal, F. P., Miraldo, S., Labrincha, J. A., & De Brito, J. (2012). An overview on concrete carbonation in the context of eco-efficient construction: Evaluation, use of SCMs and/or RAC. Construction and Building Materials36, 141-150.

Author Response

(The authors gave the same response as above.)

Round 2

Reviewer 1 Report

The Authors have improved the manuscript and they have responded to my suggestions.